# Combination of CT-Guided Microwave Ablation and Cementoplasty as a Minimally Invasive Limb-Sparing Approach in a Dog with Appendicular Osteosarcoma

**DOI:** 10.3390/ani13243804

**Published:** 2023-12-09

**Authors:** David Sayag, David Jacques, Florence Thierry, Yoann Castell, Marcel Aumann, Olivier Gauthier, Vincent Wavreille, Lambros Tselikas

**Affiliations:** 1ONCOnseil, Unité D’expertise en Oncologie Vétérinaire, 31200 Toulouse, France; 2Clinique Vétérinaire Occitanie, 31200 Toulouse, France; 3MEPY Système, 31240 Saint-Jean, France; y.castell@mepysysteme.com; 4Unité Medecine Interne, Urgences et Soins Intensifs, École Nationale Vétérinaire de Toulouse, 31300 Toulouse, France; 5Unité de Chirurgie, Anesthésie, ONIRIS—École Nationale Vétérinaire de Nantes, 44300 Nantes, France; olivier.gauthier@oniris-nantes.fr; 6Vetspecialistes, Service de Chirurgie, 1218 Grand-Saconnex, Switzerland; vwavreille@gmail.com; 7Service de Radiologie Interventionnelle, Gustave Roussy Cancer Campus, 94805 Villejuif, France; lambros.tselikas@gmail.com

**Keywords:** ablation, bone tumors, canine, heat, limb-sparing, microwave

## Abstract

**Simple Summary:**

Amputation and chemotherapy with carboplatin is the most frequently recommended standard of care to manage appendicular osteosarcoma in dogs. However, amputation is often declined by owners, or medically contraindicated. A dog with stage-I appendicular osteosarcoma was treated by a new technique allowing local control of the cancer: microwave ablation and cementoplasty, in association with a systemic chemotherapy and immunotherapy. The dog had a good quality of life after the treatment and lived 541 days after the initial diagnosis. This report illustrates for the first time a new multimodal treatment of stage-I appendicular osteosarcoma in dogs.

**Abstract:**

Image-guided microwave ablation and cementoplasty are minimally invasive techniques that have been used as part of a limb-sparing approach in the treatment of appendicular bone tumors in humans. The objective of this case report was to describe the feasibility and result of microwave ablation (MWA) and cementoplasty in a dog with stage-1 osteoblastic appendicular osteosarcoma of the right distal radius. A microwave antenna was inserted in the osteolytic area using computed tomography (CT) guidance. Three ablation cycles of 5 min at 60 watts were performed. Immediately after the MWA procedure, a tricalcium phosphate-based cement was injected through the bone trocar to consolidate the ablated zone. Adjuvant chemotherapy with six sessions of carboplatin was performed, without major complication. Response to the treatment was evaluated according to RECIST criteria every 6 weeks. Twenty-four hours after MWA, the dog was pain-free and had excellent mobility. Based on CT measurements, a reduction of the size of the lytic area was observed at the 2-month and at the 7-month follow-up (from 13% to 25% of the longest diameter), classified as stable disease according to RECIST criteria. The dog died 18 months after the initial diagnosis due to distant metastases.

## 1. Introduction

Appendicular osteosarcoma (OSA) is a common and highly malignant bone tumor in dogs [1,2]. The combination of local control, mainly by amputation of the affected limb, and adjuvant carboplatin-based chemotherapy remains the standard of care, offering improved survival rates and enhanced quality of life in affected dogs [3]. 

Amputation is often declined by owners [1,3,4,5,6]. Alternative local control techniques such as limb-sparing surgery or radiation therapy are complex and challenging. They harbor several limits. Limb-sparing surgery is associated with a lack of accessibility related to the location of the tumor, long recovery time, and high complication rates, whereas stereotactic radiation therapy is limited by technical availability in Europe [4,6,7]. 

Microwave ablation (MWA) is an interventional radiology technique that has been used in people in the treatment of various bone tumors, including both primary and metastatic lesions [6,7,8,9,10,11,12,13]. MWA involves the use of microwave energy to heat and ablate tumor cells and to destroy tumor blood vessels leading to necrosis and tumor shrinkage. The effect is both local on the tumor and its environment but also systemic by promoting anti-tumor immune response via abscopal effect [6,8,9]. This minimally invasive procedure is performed under image guidance, ideally under computed tomography (CT), to ensure the accurate and precise treatment of the tumor. 

MWA has been reported in the management of numerous cases of osteosarcoma in people, as part of a limb-sparing approach [6,10,11,12,13]. MWA can potentially reduce the extent of surgery needed to achieve local tumor control and preserve the function of the affected limb [6,13]. 

Furthermore, cementoplasty used in combination with MWA offers an additional minimally invasive option for bone tumor control and pain relief. It can be applied after MWA to increase bone stability by restoring part of its integrity and relieve pain [8,9]. 

The objective of this case report is to describe the feasibility and the clinical outcome of the combination of MWA and cementoplasty to control a stage-1 osteoblastic appendicular osteosarcoma in a dog. 

## 2. Materials and Methods

### 2.1. Owners’ Informed Consent

The method applied in this animal was based on those applied in human patients. The owners signed a complete informed consent form and allowed the publication of this case report. 

### 2.2. Case Description

A 10-year-old male Rottweiler dog presented with a stage-1 appendicular osteoblastic osteosarcoma of the right distal radius [1]. The diagnosis was based on histopathological examination following CT-guided bone biopsies. At the time of presentation, the dog presented a non-weight-bearing lameness, severe pain on palpation but no systemic abnormalities. 

A complete blood count, biochemistry profile, and urinalysis did not reveal any abnormalities. In particular, the plasma alkaline phosphatase concentration was 54 IU/L (normal range 23–212) and total calcium 111 mg/L (normal range 79–120). 

Because the owners declined amputation, microwave ablation and cementoplasty were offered to obtain local control of the neoplastic disease and achieve pain relief. 

A CT scan was acquired with 1 mm slice thickness (Canon Aquilion 80-row multi-slice CT scanner). A monostotic aggressive lytic lesion of the right distal radius (length 8.1 cm; volume of the lytic lesion: 33.7 cm^3^) associated with a periosteal reaction was identified. No distant lesions were observed. 

Multiplanar reconstructions of the right radius were performed to evaluate the area of clinical interest (exact location, length, and thickness), and to plan the MWA/cementoplasty procedure (the point of insertion of the MWA antenna, angle of insertion, and length of the canula) (Figure 1).

### 2.3. Microwave Ablation

A Saberwave ECO-200G generator (frequency 2.45 GHz; ECO Microwave) was used for the MWA procedure. The entire procedure was performed under general anesthesia (induction: diazepam 0.2 mg/kg IV and propofol 4 mg/kg IV; maintenance: isoflurane). The analgesic plan included meloxicam 0.1 mg/kg IV q24 h and fentanyl 2–4 µg/kg/min in an IV constant rate infusion during 24 h. Perioperative antibiotic prophylaxis was performed with cefalexin 15 mg/kg IV, once. 

A surgical approach of the area of clinical interest was performed by a board-certified surgeon (D.J.) to avoid tendons and other joint structure damage.

The right distal antebrachium was surgically approached via a 3 cm dorsal skin incision followed by a blunt dissection of the soft tissues. An 8-gauge Jamshidi needle (Becton-Dickinson BD, Grenoble, France) was introduced within the tumor from distal to proximal, starting within the extensor groove of the distal radius (the common digital extensor tendon was retracted laterally), avoiding the radio-carpal joint, and centered within the radial metaphysis and diaphysis (Figure 2).

A 14-gauge MWA antenna (ceramic, 200 mm, ECO system) was then coaxially inserted within the tumor using CT guidance. The ablation zone was determined by considering 2 mm beyond the limits of the area of clinical interest, as recommended in ablative protocols in humans [13]. The ablation power and time were initially determined via a comparative approach with an interventional radiologist, according to the size of the lesion and proximity of adjacent tissue on CT images (Figure 3). Three ablation cycles of 5 min (60 watts) were performed in a 2 cm spaced area. 

### 2.4. Cementoplasty

Cementoplasty was performed after the MWA, using a synthetic calcium-phosphate bone substitute (BIOCERA-VET-OSA, Theravet, Belgium). Bone cement (12 mL in total) was slowly injected within the ablation site in small batches through the Jamshidi needle (BD) to ensure the adequate filing of the excised bone area. Sequential CT acquisitions were performed during and immediately after injection to monitor the filling and distribution of the bone cement and to detect any potential leakage. 

### 2.5. Adjuvant Treatments and Analgesia

Adjuvant treatment with 6 sessions of carboplatin (300 mg/m^2^, IV, q3 weeks) and autologous vaccine as previously reported [14] (8 injections, SC—4 times weekly then 4 times monthly; APAVAC, Hastim, France) were performed following the MWA and cementoplasty procedure. 

Analgesia was obtained with monthly administration of zoledronate (4 mg/dog, IV, q4 weeks), in addition to non-steroidal anti-inflammatory drugs (meloxicam 0.1 mg/kg q24 h, orally) during all follow-ups. Tramadol (3 mg/kg q8 h, orally) and gabapentin (10 mg/kg q12 h, orally) were initially prescribed, but quickly (<3 weeks) stopped because of the excellent quality of life, and as part of the medication de-escalation. 

### 2.6. Evaluation of the Treatment Response and Quality of Life

The main criteria to evaluate the global treatment response was survival time and the score of quality of life (QOL) according to the HHHHHMM scale [15]. To evaluate the local control of the tumor, follow-up CTs of the right distal radius and thorax were performed at 2 months and 7 months after the MWA, and response was evaluated according to the response evaluation criteria in solid tumors (RECIST) by measuring the longest diameter of the lytic area [16]. CT measurements of different parameters (longest diameter, volume of dense tissue-bone and cement included, volume of cortical bone) were performed initially (D0), 2 months (D60), and 7 months (D210) after treatment by a single board-certified veterinary radiologist (F.T.). 

Radiographs of the right limb were also performed initially, 2 months, 4 months, 8 months, 10 months, 12 months, and 16 months later to monitor the evolution of the lytic lesion by avoiding repeated anesthesia. Additionally, chest radiographs were performed 10 months and 16 months after the diagnosis. 

The HHHHHMM score of QOL was evaluated every week by the owners, and every month during each clinical monitoring, by the same veterinarian (D.S.). 

## 3. Results

MWA and cementoplasty were performed without any complication. Initially, the score for quality of life was 33/70 points according to the HHHHHMM scale [14]. This score markedly improved and was 52/70 points 24 h after treatment. Pain medications were tapered off, and only NSAIDs (meloxicam 0.1 mg/kg q24 h, orally) were continued. Adjuvant chemotherapy and immunotherapy started 5 days after the procedure. No significant side effect of the treatment was reported.

CT measurements of different parameters after treatment are reported in Table 1. Seven months after the MWA and cementoplasty, a reduction of 25% of the longest diameter of the lytic area was observed, corresponding to a stable disease according to RECIST. 

Three months after treatment, the dog presented for an acute grade 3/6 lameness. A radial fracture line extending to the lytic area was noted. A Robert Jones splint dressing was applied, and the dog’s quality of life remained good (HHHHHMM score > 50/70) despite of the conservative management of the pathological fracture. The fracture did not heal during the follow-ups, and the dog kept the Robert Jones splint dressing during all his life, because the owners declined any surgical stabilization (Figure 4).

Seven months post treatment, osteomyelitis due to *Staphylococcus pseudintermedius* was identified after bacterial analysis of a fine-needle aspiration of lytic tissue. This infection could explain the overall loss of bone density at D210. A 6-week course of antibiotic therapy chosen according to the result of the bacterial analysis (cefalexin, 15 mg/kg/q12 h, orally) allowed a complete recovery. 

Ten months post treatment, three pulmonary nodules (between 1 and 2.5 cm in diameter) were identified on thoracic radiographs. Toceranib phosphate (2.4 mg/kg every other day, orally) was prescribed, but quickly (less than 2 weeks after the prescription) suspended due to major gastrointestinal side effects. From that moment, only painkillers (meloxicam 0.1 mg/kg q24 h orally; zoledronate 4 mg/dog q4 weeks IV) were continued. 

The dog died 541 days after the initial diagnosis due to the progression of the metastatic disease.

## 4. Discussion

This report illustrates for the first time the clinical application of the combination of MWA and cementoplasty as a limb-sparing approach in a multimodal treatment of a dog with appendicular osteosarcoma. A local stable disease and long survival time were observed. 

In human medicine, MWA has been used in the treatment of bone tumors for more than 30 years. Recent technical improvements in MWA devices allow the creation of more accurate ablation zones: an optimal local control for up to 75% of cases with tumors less than 3 cm has been described in a couple of recent studies [8,9]. MWA can be used as an independent percutaneous treatment or as an ancillary treatment for hemostasis, tumor inactivation, or to improve the safety of tumor resection margins. In recent years, there has been a growing interest in ablative therapies for the treatment of solid tumors in people, leading to the recent publication of guidelines [9]. In a retrospective study including 15 patients who underwent MWA for recurrent bone tumors, Zheng et al. reported effective postoperative pain relief and interesting local control rates [13].

In a larger retrospective study including 79 patients with osteosarcoma of the distal tibia, no significant difference in survival, local recurrence rate, or major complication rate between MWA and amputation was observed. Moreover, a functional advantage in favor of MWA was reported [10]. MWA could also allow joint-sparing surgery in patients with osteosarcoma of the proximal tibia [11]. 

In dogs, two studies evaluated the feasibility, biomechanical properties, and efficacy of MWA for the local treatment of distal radial osteosarcoma [17,18]. Moreover, the management of pulmonary metastasis of OSA using MWA has been reported in three cases [19,20]. In the pilot study conducted by Salyer and others [18], six dogs with a cytological diagnosis of distal radius osteosarcoma underwent fluoroscopy-guided MWA 48 h before amputation. After amputation the histopathological analysis revealed a median tumor necrosis rate of 55% (30% to 90%). This pilot study demonstrated the short-term efficacy of MWA in canine osteosarcoma tissue.

Moreover, a second study by Kalamaras and colleagues [17] did not demonstrate any significant modification in the biomechanical properties of normal canine distal radius after MWA. Based on these preclinical data, MWA was decided in the case reported herein. 

Percutaneous cementoplasty with polymethylmethacrylate bone cement was reported as part of a multimodal treatment of primary bone tumors of the distal aspect of the radius in a pilot study involving four dogs [21]. A significant improvement of lameness was reported. Complications included deep wound infection, intraarticular cement leakage, and venous thrombosis. More recently, a new self-setting bone substitute was associated with a decrease in pain associated with bone cancer, and an improvement in quality of life [22].

In this reported case, we used calcium phosphate cement. Calcium phosphates are biocompatible, osteoconductive, and bioresorbable [23,24,25]. They offer an intrinsic microporous structure for the transport of nutrients and metabolic products. Although a great number of intrinsic variabilities are described in the literature, several studies reported that PMMA and oil-based calcium phosphate cement had similar biomechanical performance [25]. In the reported case, osteomyelitis was observed 7 months after cementoplasty, in the context of a persistent pathological fracture. The pieces of bone cement within the lytic area may have favored bacterial proliferation and the development of osteomyelitis. 

The main complication in the management of the reported case was the appearance of a pathological fracture three months after the procedure despite cementoplasty. In this situation, cementoplasty was not sufficient to counteract the biomechanical constraints associated with MWA. In human guidelines, appropriate internal fixation associated with cementoplasty is recommended after the MWA of bone tumors in extremities by using bone plates or intramedullary nails [9]. 

After this first case, our team performed MWA as part of a multimodal management of appendicular osteosarcoma in three additional dogs, with post-procedure stabilization of the treated site using cementoplasty, internal fixation, +/− bone graft. No pathological fracture was reported in these three cases. Two dogs had osteosarcoma of the distal radius, and one dog had osteosarcoma of the proximal ulna. The quality of life was significantly improved in two out of the three dogs after the MWA, cementoplasty, and surgical stabilization. All dogs received the same immuno-chemotherapy protocol. One dog died from metastatic disease 4 months after the initial diagnosis. The two other dogs are still alive at the time of the manuscript submission (respectively, 4 months and 1 months after the MWA). One dog did not have significant improvement of his QOL in the weeks following MWA, and amputation was performed 2 months after the MWA. 

QOL remains a multidimensional concept that involves a subjective evaluation of factors that contribute to overall well-being. The evaluation of QOL is the cornerstone in therapeutic decision making for cancer-bearing patients. The objective of new treatment modalities and treatment approach is to offer better control of the cancer, and a longer survival with an optimal QOL [26]. QOL could be evaluated according to different scales. In this case report, we choose to evaluate QOL according to the HHHHHMM scale, which evaluates hurt, hunger, hydration, hygiene, happiness, and mobility. The HHHHHMM is a numerical score that allowed evaluation by the owners, and adequate monitoring by the veterinarian. Initially, the patient had a decrease in his QOL (score 32/70), but this score rapidly improved after the MWA and cementoplasty, mainly due to adequate pain control. According to the owners, the dog had an adequate QOL during his follow-up, and the QOL score remained superior to 50/70 up to 1 week before the dog’s death. The medical evaluation of the QOL was always performed by the same veterinarian to avoid bias. 

## 5. Conclusions

The combination of MWA and cementoplasty may represent a new limb-sparing option, as part of a multimodal management of appendicular osteosarcoma in dogs. Technical improvements and more patients are needed to evaluate the efficacy of local control and safety in comparison to amputation and other limb-sparing procedures. 

## Figures and Tables

**Figure 1 animals-13-03804-f001:**
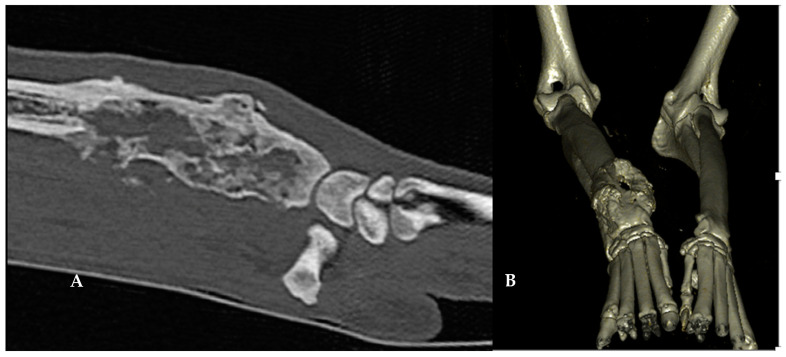
Sagittal (**A**) and 3D CT reconstructions (**B**) of the right distal radius of a 10-year-old dog with stage-1 appendicular osteosarcoma.

**Figure 2 animals-13-03804-f002:**
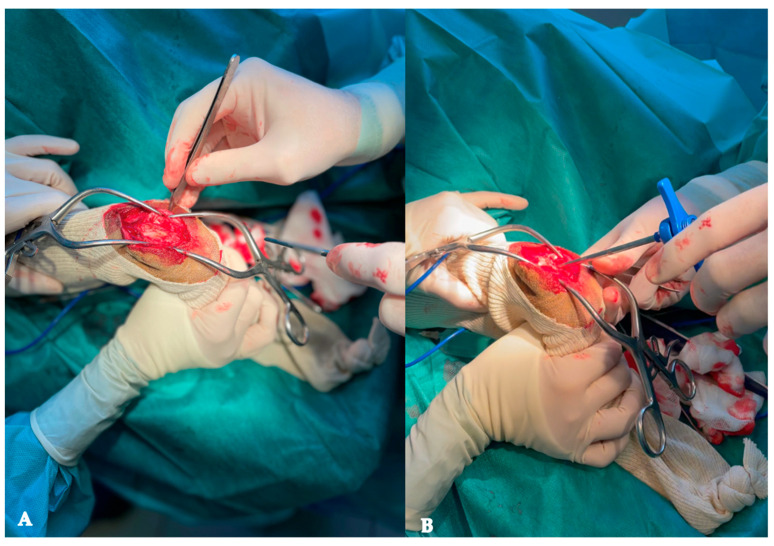
Pictures of the surgical approach of the area of clinical interest in a dog with right distal radius osteosarcoma, (**A**) after the retraction of the tendons and exposure of the extensor groove, and (**B**) after the introduction of the Jamshidi needle within the tumor.

**Figure 3 animals-13-03804-f003:**
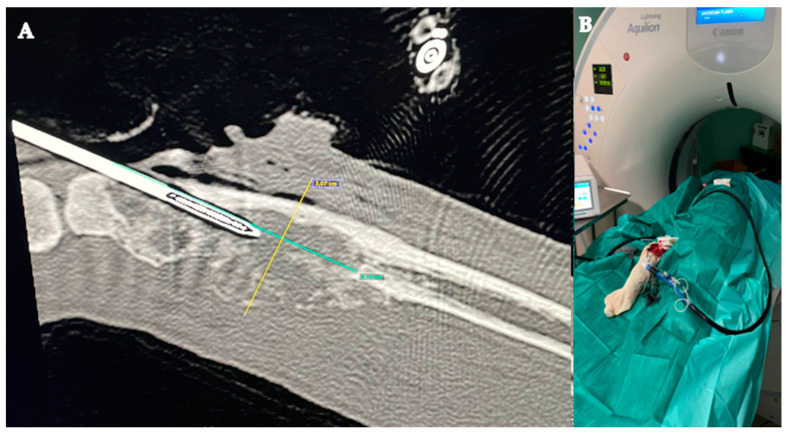
(**A**) Intraoperative CT image during the microwave ablation. (**B**) Positioning of the dog within the CT machine during the microwave ablation.

**Figure 4 animals-13-03804-f004:**
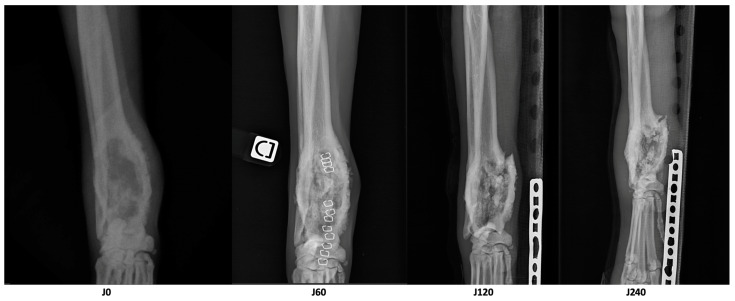
X-rays of the right distal radius in a dog with appendicular osteosarcoma treated using microwave thermal ablation (MWA), cementoplasty, and immuno-chemotherapy. The X-rays were performed initially (J0), 2 months (J60), 4 months (J120), and 8 months (J240) after the MWA. A pathological fracture was diagnosed 3 months after the MWA, and persisted up to the end of the life of the patient.

**Table 1 animals-13-03804-t001:** Measurements using the CT images performed before treatment, at 2 months (D60), and 7 months (D210) after microwave ablation of an appendicular osteosarcoma in a dog.

Parameter	D0	D60	D210	Variation D60/D0	Variation D210/D0
Longest diameter of the osteolytic area (cm)	8.1	6.7	6.1	−17%	−25%
Volume of dense tissue in the osteolytic area (cm^3^)	18.8	21.1	17.5	+12%	−7%
Volume of the cortical bone of the area of interest (cm^3^)	8.2	10.6	7.7	+29%	−6%

## Data Availability

The data presented in this study are available in the article.

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
