# Peer review of "Combination of CT-Guided Microwave Ablation and Cementoplasty as a Minimally Invasive Limb-Sparing Approach in a Dog with Appendicular Osteosarcoma"

_animals, 2023, doi:10.3390/ani13243804_

Round 1

Reviewer 1 Report

Comments and Suggestions for Authors

Dear Authors:

 I was asked to revise your case study “Combination of CT-guided microwave ablation and cementoplasty as a minimally invasive limb-sparing approach in a dog with appendicular osteosarcoma.”

While I believe it is of good value for the veterinary community, given the novelty of the pathology therapeutical approach, I think it must be improved.

That said, I recommend its publication after some revisions.

When revising your manuscript, carefully consider all issues mentioned in my comments: outline each change made in response to my comments and provide adequate rebuttals for any comments not covered.

Kind regards,

The Reviewer

Comments on the Quality of English Language

Some minor spelling issues were detected. Please correct those highlighted and carefully review the document for more.

Author Response

Thank you very much for taking the time to review our manuscript.

We improved our manuscript in order to fully answer to all your recommendations. 

Please find in attachment our complete answer. 

Best regards

Reviewer 2 Report

Comments and Suggestions for Authors

Thank you for the submission of this case report. It is very interesting and I hope these methods can be used in a larger case series or prospective clinical trial.

It is disappointing that the follow up CTs were only of the only the limb and not inclusive of the thorax. I would strongly recommend including the thorax in the CT of any patient with an appendicular osteosarcoma.

I appreciate the end of the discussion where the authors addressed the early pathologic fracture and consider internal fixation for future cases.

I recommend separating limits by treatment in lines 47-49. Ex limits of limb-sparing surgery in a different sentence from the limits of radiation therapy as the limits are very different. The current sentence is confusing.

Please add a reference for the staging system used to assign this case a stage 1.

On line 114- please add the reference for the ablative protocols in humans.

Please list the route of administration of the autologous vaccine on line 130 and meloxicam on line 133

Can the authors please provide a reference or more information about the autologous vaccine that was used?

Line 147- should this have ref 14 not 16?

Please add route of administration of antibiotic in line 165; same of toceranib on line 167

Comments on the Quality of English Language

Several grammatical errors and inappropriate punctuation are noted throughout this publication including in the description text of some of the figures.

Author Response

Thank you very much for taking the time to review our manuscript.

We improved our manuscript in order to fully answer to all your recommendations. 

1/ About the follow-ups CT

All follow-ups CT included thoracic evaluation. CTs were evaluated by the same ECVDI board certified veterinarian. We completed the manuscript. 

2/ About the future cases 

We completed this part of the discussion, in view of the most recent follow-up in other patients. At this time, 3 additional dogs with appendicular osteosarcoma were treated by MWA, cementoplasty, and immuno-chemotherapy. 

3/ About introduction

As you recommended, we reformulate this part of the introduction. 

We added reference to the staging system. 

4/ Reference in M&M

In this case, the MWA protocol was determined by multi-disciplinary (oncologist, surgeon, radiologist) and transversal approach with human interventional radiologist. We added reference to MWA in human patients with bone tumor. 

5/ About autologous vaccine

Autologous vaccination is available to manage several tumors in dogs, but level of evidence in literature remains very low at this time. The treatment is homologated in dogs for any tumor except T-cell lymphoma. We added a reference about the use of this technic in dogs with osteosarcoma. 

6/ Route of administration

We added route of administration for all the treatment used. 

Round 2

Reviewer 1 Report

Comments and Suggestions for Authors

30th November 2023

Dear Authors:

Thank you for your great effort in responding to the previous revision.

I believe your paper is ready for publication after some minor revisions (some typos).

Please find attached the file with the specified revisions.

Best regards,

The reviewer.

Comments on the Quality of English Language

The document has some minor English spelling typos. The authors must opt for British English (BE) or American English (AE), and correct the whole document for double spellings. Some examples are the words tumour (BE), harbor (AE), and anesthesia (AE). 

Author Response

Dear reviewer,

We would like to thank you again for taking the time to review our manuscript.

We modified our manuscript to fully answer to all your recommendations.

Yours Sincerely,

The authors